# UTBoost: A Tree-boosting based System for Uplift Modeling

## Abstract

Uplift modeling comprises a collection of machine learning techniques designed for managers to predict the incremental impact of specific actions on customer outcomes. These models enable decision-makers to identify customer segments most likely to exhibit positive responses to targeted interventions, thus optimizing resource allocation and maximizing overall returns. However, accurately estimating this incremental impact poses significant challenges due to the necessity of determining the difference between two mutually exclusive outcomes for each individual. In our study, we introduce two novel modifications to the established Gradient Boosting Decision Trees (GBDT) technique. These modifications sequentially learn the causal effect, addressing the counterfactual dilemma. Each modification innovates upon the existing technique in terms of the ensemble learning method and the learning objective, respectively. Experiments with large-scale datasets validate the effectiveness of our methods, consistently achieving substantial improvements over baseline models.

## 1 Introduction

Uplift modeling, a machine learning technique used to estimate the net effect of a particular action, has recently drawn considerable attention. In contrast to traditional supervised learning, uplift models concentrate on modeling the effect of a treatment on individual outcomes and generate uplift scores that show the probability of persuasion for each instance. This technique has proven particularly useful in personalized medicine, performance marketing, and social science. These scenarios are typically presented using tabular data, with each entity having an indicator variable representing whether the subject was treated or not. Additionally, each entity includes a corresponding label.

However, a notable challenge in uplift modeling is the absence of observations for both treated and control conditions for an individual within the same context, complicating the direct application of standard classification techniques for modeling individual treatment effects. Various tree-based studies have tackled this challenge by creating measures of outcome differences between treated and untreated observations and maximizing heterogeneity between groups (Rzepakowski & Jaroszewicz, 2012; Athey & Imbens, 2016). Several research studies provide further generalization to bagging ensemble methods on the idea of random forests (Guelman et al., 2012; 2015), which aim to address the challenges of tree model performance decay with an increasing number of covariates. Despite the success of these nonparametric methods in prediction, we have experimentally discovered that random-forest based methods still suffer from significant degradation of power in predicting causal effects as the number of variables increases.

Boosted trees present an alternative, where subsequent trees are fitted based on the causal effects identified by preceding models. Our findings indicate that while both ensemble learning methods perform comparably in low-dimensional settings, boosting exhibits significant advantages in handling high-dimensional data.

However, the criterion for maximizing heterogeneity often overlooks outcome learning, focusing solely on uplift signals. This limitation affects prediction accuracy and restricts the models' applicability in areas where accurate outcome prediction is essential. For instance, in performance marketing, to ensure campaign

fairness and profitability, managers must consider both the natural probability of conversion and the predicted incremental score, preventing high-propensity customers from being overlooked.

Recent advancements have explored solutions from a potential outcomes framework, with meta-learning, particularly, gaining traction for its versatility in employing any machine learning estimator as a base model. The Single-Model (Athey & Imbens, 2015) and Two-Model (Radcliffe, 2007) approaches represent two prevalent meta-learning strategies. The first method is trained over an entire space, with the treatment indicator serving as an additive input feature. However, this method's drawback is that a model may not select the treatment indicator if it only uses a subset of features for predictions, such as a tree model. Consequently, the causal effect is estimated as zero for all subjects. An enhancement over the Single-Model method involves using two separate models to represent the two potential outcomes. However, this dual-model approach does not leverage the shared information between control and treated subjects, resulting in cumulative errors.

Recently, neural network-based methods have become mainstream in uplift modeling. TARNet (Shalit et al., 2017) estimates causal effects by learning two functions through neural networks, with a shared feature representation for treated and control subjects. This architecture overcomes the limitations of the Two-Model approach. Nonetheless, TARNet and similar methods do not explicitly incorporate the heterogeneity of causal effects into the training loss objective, leading to diminished performance in scenarios with weak causal effects and similar label distributions across groups.

In this paper, we propose CausalGBM (Causal Gradient Boosting Machine), a new nonparametric method that utilizes tree models as base learners to simultaneously learn causal effects and potential outcomes through loss optimization. Our method explicitly computes the contribution of causal effects to the loss function at each split selection, avoiding the defect that the treatment indicator may not be picked up in some scenarios where the causal effects are weak. We further enhance the model's convergence speed by incorporating second-order gradient information. Experimental comparisons on four datasets demonstrate that our model outperforms baseline methods and shows better robustness.

With the increasing reliance on automated A/B testing platforms and the consequent data volume expansion, there is a pressing need for more computationally efficient models. In the context of randomized trials, we introduce a multi-objective approximation method to improve CausalGBM's computational efficiency and scalability. This method simplifies the binary quadratic optimization problem into a series of univariate quadratic problems. We have implemented our novel techniques in the UTBoost (Uplift Tree Boosting system) software, available under the MIT license.

We highlight our contributions as follows:

1. We innovate the uplift tree method that focuses on maximizing heterogeneity by extending the ensemble of trees from bagging to boosting.

2. For the first time, we integrate potential outcomes and causal effects within the classical GBDT framework, employing a second-order method to fit the multi-objective function.

3. In the context of randomized trials, we propose an approximation algorithm that significantly reduces the computational complexity of the CausalGBM algorithm.

4. Through extensive experimentation on real-world and public datasets, we demonstrate that our models outperform baseline methods and exhibit superior robustness.

The rest of the paper is organized as follows. In Section 2, we review the existing literature on uplift modeling. Section 3 introduces the causal inference framework and notations. Sections 4 and 5 explain the uplift boosting algorithms. In Section 6, we describe the experiments and report the results. Finally, Section 7 presents our conclusions.

## 2  Related Work

Existing work on uplift modeling can be categorized into various frameworks. One common framework is meta-learning, which utilizes various machine learning estimators as base learners to estimate the treatment

effect (Athey & Imbens, 2015; Radcliffe, 2007; Jaskowski & Jaroszewicz, 2012). Notable examples of meta-learning paradigms include Single-Model and Two-Model approaches. However, when the sizes of the two groups are uneven, the models may perform inconsistently, which could undermine the accuracy of the causal effect estimates. To address this issue, the X-learner was proposed, which leverages information learned from the control group to improve the estimation of the treated group and vice versa (Künzel et al., 2019).

In the realm of neural networks, representation learning has become a key strategy for addressing biases in treatment distribution, a technique that is also applied in uplift modeling. These methods utilize networks with layers that are either specific to each group or shared between them. An example is the Balancing Neural Network (BNN) (Johansson et al., 2016), which aims to reduce the differences between the treatment and control group distributions by applying an Integral Probability Metric (IPM). Building on the BNN, the Treatment-Agnostic Representation Network (TARNet) and Counterfactual Regression (CFR) (Shalit et al., 2017) use a dual-headed structure to facilitate multi-task learning, developing two functions from a common, balanced representation of features.

Tree-based methods have also attracted considerable attention. Rzepakowski & Jaroszewicz (2012) put forward decision tree-based methods for uplift modeling that use one of the following splitting criteria: Kullback-Leibler (KL) divergence, squared Euclidean distance (ED), and chi-squared divergence. The authors compare the KL and ED-based models to a few baseline approaches, including the Two Model method, and found that their proposed methods outperform the baselines in the binary-treatment case. Athey & Imbens (2016) introduced the causal tree algorithm, which relies on a criterion centered on treatment effects for splitting. Moving forward, Rafla et al. (2023) proposed a Bayesian criterion for evaluating uplift decision trees, with the goal of improving their generalizability and reducing the risk of overfitting.

Furthermore, advancements in ensemble methods have improved model performance. Many of these methods use tree-based models as their core learners. Inspired by traditional random forests, Guelman et al. (2015) introduced an uplift random forest framework. Similarly, Wager & Athey (2018) presented a causal forest algorithm that estimates treatment effects using causal trees as the base learner. An alternative to random forests is boosted trees. This type of methods builds up a function approximation by successively fitting weak learners to the residuals of the model at each step. Powers et al. (2018) developed a causal boosting algorithm that employed causal trees as base learners, estimating treatment-specific means rather than treatment effects. This algorithm updates the residuals for each sample in every iteration, which are then used to train new models. However, recalculating residuals for the entire sample set can carry over biases from previous iterations, potentially leading to compounded errors in subsequent models. We refine this process by limiting residual calculations to the treated group only, reducing dependencies between learners and ensuring that subsequent models have the necessary information to adjust for bias. Furthermore, we have optimized the loss function, extending the reach of the standard gradient boosting algorithm into causal effect estimation and thus narrowing the divide between these two methodologies.

## 3 Notation and assumptions

Uplift modeling seeks to quantify the incremental effect of interventions on individual outcomes. Given $n$ independent and identically distributed samples $\{(\mathbf{X}_i, y_i, w_i)\}_{i=1}^n$, where each sample comprises $p$ features $\mathbf{X}_i \in \mathbb{R}^p$, an observed outcome $y_i \in \mathbb{R}$, and a binary treatment indicator $w_i$, which indicates whether unit $i$ received treatment ($w_i = 1$) or control ($w_i = 0$), the objective is to predict the net effect of the treatment (Gubela et al., 2019; Devriendt et al., 2018). In the uplift modeling literature (Zhang et al., 2021), it is typical to assume that the treatment $w_i$ is randomly assigned and we commonly take the following quantity as the learning objective:

$$\tau(\mathbf{x}) = \mathbb{E}[y_i | w_i = 1, \mathbf{X}_i = \mathbf{x}] - \mathbb{E}[y_i | w_i = 0, \mathbf{X}_i = \mathbf{x}], \tag{1}$$

which signifies the uplift on $y$ caused by the treatment $w$ for the subject with feature $\mathbf{x}$.

It should be noted that although $\tau(\mathbf{x})$ is interpreted from a causal perspective, equation 1 is not defined within a causal framework. In the following, we briefly show that under certain assumptions, estimating the uplift $\tau(\mathbf{x})$ is equal to inferring the individual treatment effect, a causal quantity defined in the potential outcomes framework (Neyman, 1923; Rubin, 1974). Let $y_i(1)$ and $y_i(0)$ be the potential outcomes that

would be observed for unit $i$ when $w_i = 1$ and $0$, respectively. The individual treatment effect (ITE) is defined as

$$\tau_i := y_i(1) - y_i(0), \tag{2}$$

and $\mathbb{E}[y_i(1) - y_i(0)|\mathbf{X}_i]$ is the best estimator in terms of the mean squared error (Künzel et al., 2019). In the causal inference literature (Imbens & Rubin, 2015; Imbens & Wooldridge, 2009), the following assumptions (Assumptions 1-2) are commonly made.

**Assumption 1 (Consistency)** $y_i = y_i(1)w_i + y_i(0)(1 - w_i)$

**Assumption 2 (Ignorability)** $\{y_i(1), y_i(0)\} \perp\!\!\!\perp w_i|\mathbf{X}_i$

These assumptions ensure that the expected outcomes under treatment and control conditions can be accurately estimated from the observed data. Under Assumptions 1-2, we have

$$\mathbb{E}[y(w)|\mathbf{X}] \overset{\text{Asp 1}}{=} \mathbb{E}[y(w)|\mathbf{X}, w] = \mathbb{E}[y(1)w + y(0)(1 - w)|\mathbf{X}, w] \overset{\text{Asp 2}}{=} \mathbb{E}[y|\mathbf{X}, w],$$

where the 1st and 3rd equation relies on Assumption 1 and 2, respectively, and the 2nd equation can be verified by taking $w = 1, 0$. Then we have $\mathbb{E}[y(1) - y(0)|\mathbf{X}] = \mathbb{E}[y|\mathbf{X}, w = 1] - \mathbb{E}[y|\mathbf{X}, w = 0]$.

To summarize, uplift modeling for $\tau(\mathbf{x})$ in equation 1 is equivalent to estimating the ITE defined in equation 2 under Assumptions 1-2. Further, if we replace the condition $\{\mathbf{X} = \mathbf{x}\}$ in equation 1 to a more rough granularity such as $\{\mathbf{X} \in \mathbf{X}_t\}$ (*e.g.*, restricting the sample in a leaf node $t$), then the equivalence between uplift modeling and ITE estimation requires $w$ to be randomly assigned, which is a stronger condition than the ignorability assumption (Assumption 1). In the following, we focus on the setting with randomly assigned treatment.

## 4    Tree Boosting for Treatment Effect Estimation

Our first proposed method adopts a sequential learning approach to fit causal effects without directly modeling potential outcomes. As the splitting criterion in the training process is similar to the standard delta-delta-p (DDP) algorithm (Hansotia & Rukstales, 2002), which aims to maximize the difference of causal effect between the left and right child nodes, once the boosting method is overlooked. We refer to this as TDDP (Transformed DDP).

### 4.1    Ensemble Learning with Transformed Labels

Taking the decision tree as the base learner, we use a sequence of decision trees to predict the uplift $\tau(\mathbf{x})$. As the uplift cannot be observed for each sample unit, the ensemble method for $\tau(\mathbf{x})$ differs from the common supervised-learning scenarios. We explicitly derive the optimization target for uplift estimation in each iteration of the boosting.

Denoting a tree model by $T(\mathbf{x}; \theta_j)$, where $\theta_j$ encapsulates the model's parameters, including partitioning and leaf node estimation, our goal is to predict the uplift through the following equation:

$$\widehat{\tau}(\mathbf{x}) = \sum_{j=1}^{M} T(\mathbf{x}; \theta_j). \tag{3}$$

We sequentially optimize $T(\mathbf{x}; \theta_j)$ to minimize the loss associated with $\widehat{\tau}(\mathbf{x})$.

Let $u_m = \sum_{j=1}^{m} T(\mathbf{x}; \theta_j)$ represent the cumulative prediction of the first $m$ trees. At step $m$, with the current prediction $u_m$, the loss is denoted by $\mathcal{L}(\tau(\mathbf{x}), u_m(\mathbf{x}))$. The gradient, then, is defined as:

$$\boldsymbol{g}_m := \frac{\partial \mathcal{L}(\tau(\mathbf{x}), u_m(\mathbf{x}))}{\partial u_m(\mathbf{x})}$$

where $-\boldsymbol{g}_m$ indicates a local direction for further decreasing the loss at $u_m$. Thus, in a greedy manner, we fit $T(\mathbf{x}; \theta_{m+1})$ to approximate $\boldsymbol{g}_m$.

Specifically, for uplift modeling, the quadratic loss at step $m$ can be expressed as:

$$\mathcal{L}(\tau(\mathbf{x}), u_m(\mathbf{x})) = \frac{1}{2}\Big\{\mathbb{E}[y|\mathbf{X} = \mathbf{x}, w = 1] - \mathbb{E}[y|\mathbf{X} = \mathbf{x}, w = 0] - u_m(\mathbf{x})\Big\}^2$$

with the coefficient $\frac{1}{2}$ simplifying gradient computation. The negative gradient then becomes:

$$-\frac{\partial \mathcal{L}(\tau(\mathbf{x}), u_m(\mathbf{x}))}{\partial u_m(\mathbf{x})} = \mathbb{E}[y|\mathbf{X} = \mathbf{x}, w = 1] - \mathbb{E}[y|\mathbf{X} = \mathbf{x}, w = 0] - u_m(\mathbf{x}) = \mathbb{E}[y - u_m(\mathbf{x})|\mathbf{X} = \mathbf{x}, w = 1] - \mathbb{E}[y|\mathbf{X} = \mathbf{x}, w = 0].$$

Thus, in constructing the $(m + 1)$-th tree, $T(\mathbf{x}; \theta_{m+1})$, we transform the outcome $y_i$ to $y_i - u_m(\mathbf{X}_i)$ for treated units, while maintaining the original outcome for the control group. This approach allows for tree construction based on these transformed outcomes. Algorithm 1 outlines the overall training procedures for TDDP, where the *split criterion* temporarily serves as a placeholder and the details will be introduced in the following subsection 4.2.

---

**Algorithm 1** Gradient Tree Boosting for Uplift Modeling

---

**Input:** Data: $\mathcal{D} = \{(\mathbf{X}_i, y_i, w_i)\}_{i=1}^{N}$, Shrinkage rate: $\alpha$
**Output:** $u_M = \sum_{m=1}^{M} T(\mathbf{x}; \theta_m)$

1: Set $u_0(\mathbf{x}) = 0$.
2: **for** $m = 1, \cdots, M$ **do**
3:      Set $y_i = y_i - T(\mathbf{x}, \theta_{m-1})$ for $\{i \mid w_i = 1\}$.
4:      **Build Tree Structure:** Recursively partition $\mathcal{D}^m$:
5:      **while** the *stopping rule* is not satisfied **do**
6:          Select the optimal split $s^*$ in the candidate splits by criterion equation *split criterion*.
7:          Split the current node into child nodes by $s^*$.
8:      **end while**
9:      Output the Tree Structure $T_m$
10:      **Obtain the Estimator $T(\mathbf{x}; \theta_m)$:**
11:      **for** each leaf node $t_i$ of $T_m$ **do**
12:          Get $D^m(t_i)$: the sample units in $D^m$ that fall into $t_i$.
13:          Estimate Weight: $\widehat{\tau}_m(t_i) = \bar{Y}_1(D^m(t_i)) - \bar{Y}_0(D^m(t_i))$.
14:          Output the m-th predictor: $T(\mathbf{x}; \theta_m) = \alpha\widehat{\tau}_m(t_{T_m}(\mathbf{x}))$, where $t_{T_m}(x)$ denotes the leaf node that $x$ belongs to in $T_m$.
15:      **end for**
16: **end for**

---

## 4.2 Tree Construction Method

Here, we delve into the split criterion, a pivotal component of our tree construction methodology.

**Split Criterion** Traditional CART algorithms select splits to minimize mean squared errors (MSE) in regression trees. However, this approach is not directly applicable in uplift modeling due to the unobservability of unit-level uplift ($\tau_i$). In our context, we adapt this criterion to leverage aggregated uplift statistics, such as averages or variances, available within groups of units.

In the next, we will show that minimizing the MSE is equivalent to maximizing the gaps between the average uplift within the split nodes. Consider the split selection at an internal root node $t$ with data $D_t := \{\mathbf{X}_i, y_i, w_i\}_{i=1}^{n_t}$. Let $s$ denote a split, $s_L$ and $s_R$ denote the indices set in the left and right child nodes with sub-sample size $n_L$ and $n_R$, respectively, under the split $s$. For example, suppose $s = \{x_j = a\}$ for a numeric variable $x_j$, then $s_L = \{i|X_{ij} \leq a\}$ and $s_R = \{i|X_{ij} > a\}$. Let $\bar{\tau}_L := \sum_{i \in s_L} \frac{\tau_i}{n_L}$ and $\bar{\tau}_R := \sum_{i \in s_R} \frac{\tau_i}{n_R}$ denote the average uplift in the left and right child nodes, respectively. Then we have the following proposition:

**Proposition 1** *Minimizing the mean squared errors of $\tau_i$ in the split nodes is equivalent to maximizing the difference between the average uplift within the left and right child nodes, i.e.,*

$$argmin_s \left\{ \sum_{i|i\in s_L} \left(\tau_i - \bar{\tau}_L\right)^2 + \sum_{i|i\in s_R} \left(\tau_i - \bar{\tau}_R\right)^2 \right\} = argmax_s \left\{ \frac{n_L n_R}{n}(\bar{\tau}_L - \bar{\tau}_R)^2 \right\}.$$

Proposition 1 guides us to a practical split criterion for the uplift modeling as both $\bar{\tau}_L$ and $\bar{\tau}_R$ are aggregated values that can be estimated from data. Taking $\bar{\tau}_L$ as an example, the definition of $\bar{\tau}_L$ involves $\{y_i(1), y_i(0)\}$ as shown in equation equation 4,

$$\bar{\tau}_L = \frac{\sum_{i\in s_L} y_i(1) - y_i(0)}{n_L} = \bar{Y}_L(1) - \bar{Y}_L(0). \tag{4}$$

Under randomly assigned treatment, $\bar{Y}(1)$ and $\bar{Y}(0)$ can be estimated by the sample average of $y$ in the treatment and control groups, respectively. Therefore, the optimal split $s^*$ is selected by the following rule:

$$s^* = \arg\max_s \left\{ \frac{n_L n_R}{n} \left[ \left(\bar{Y}_L^1 - \bar{Y}_L^0\right) - \left(\bar{Y}_R^1 - \bar{Y}_R^0\right) \right]^2 \right\}, \qquad (\textit{split criterion})$$

where $Y_L^1 := \frac{\sum_{i|i\in s_L, w_i=1} y_i}{n_L^1}$ with $n_L^1$ denoting the number of treated units in the left child node, and $Y_L^0$, $Y_R^1$, $Y_R^0$ are defined similarly.

It's important to note that TDDP, while inspired by gradient boosting, diverges from it by focusing on observed outcomes and directing weak learners towards uncovering heterogeneities in treatment effects rather than strictly following a gradient descent path.

## 5 Causal Gradient Boosting Machine

The TDDP algorithm used to identify the heterogeneity of causal effects skips the fitting of potential outcomes and has limitations in scenarios where potential outcomes are still needed for decision making. Meanwhile, in order to emphasize the importance of causal individual causal effect estimation in the model, we abandon the previous framework of obtaining causal effect estimates derived from outcome prediction. We propose to use Causal Gradient Boosting Machine (CausalGBM) to fit causal effects and outcomes in a single learner. This approach extends the standard gradient boosting algorithm to the field of causal effect estimation, thus bridging the gap between the two classes of methods.

### 5.1 Learning Objective

In order to realize the simultaneous estimation of the two objectives, we split the original single learning task. Under equation 1 and equation 2, we can conduct that:

$$y_i = y_i(1)w_i + y_i(0)(1 - w_i) = w_i\tau_i + y_i(0) \tag{5}$$

which indicates that for treated instances, observed outcomes are the sum of potential outcomes and individual causal effects, while for control instances, they equate to the potential outcomes alone. This relationship facilitates the indirect learning of both potential outcomes and individual causal effects from the observed data.

We employ a tree ensemble model with $2M$ additive functions to predict the output for a dataset with $n$ samples and $p$ features:

$$\hat{y}_i = \sum_{m=1}^{M} f_m(\mathbf{X}_i) + w_i\tau_m(\mathbf{X}_i), \ f_m, \tau_m \in \mathcal{F}$$

where $\mathcal{F} = \{f(\mathbf{X}) = v_{q(\mathbf{X})}, \tau(\mathbf{X}) = u_{q(\mathbf{X})}\}(q : \mathbb{R}^p \to T, v \in \mathbb{R}^T, u \in \mathbb{R}^T)$ represents the regression trees space. Here $q$ maps each example to the corresponding leaf index in each tree, and $T$ refers to the number of leaves

in the tree. Note that leaf weights comprise both $u$ and $v$ in this framework, which is significantly different from the classical regression tree. We will use the decision rules in the trees (given by $q$) to classify instances to leaves and compute the final predictions by summing up the scores by equation 5 in the corresponding leaves (given by $u, v$). To learn the set of functions that are employed in the ensemble model, we minimize the following objective function:

$$\mathcal{L}(\theta) = \sum_i l(y_i, \hat{y}_i)$$

Here $l$ is a differentiable convex loss function that measures the difference between the prediction and the observed label. Using the binary decision tree as a meta-learner, we train the ensemble model sequentially to minimize loss. In other words, let $\hat{y}_i^t$ be the prediction of the i-th instance at the t-th iteration, we add $f_t + w_i \tau_t$ to minimize the following objective:

$$\mathcal{L}^{(t)} = \sum_{i=1}^n l(y_i, \hat{y}_i^{(t-1)} + f_t(\mathbf{X}_i) + w_i \tau_t(\mathbf{X}_i)) = \sum_{i=1}^n l(y_i, \hat{y}_i(0)^{(t-1)} + f_t(\mathbf{X}_i) + w_i(\hat{\tau}_i^{(t-1)} + \tau_t(\mathbf{X}_i)))$$

We employ a second-order approximation to expedite the optimization process.

$$\mathcal{L}^{(t)} \simeq \sum_{i=1}^n [l(y_i, \hat{y}_i^{(t-1)}) + \frac{\partial l(y_i, \hat{y}^{(t-1)})}{\partial \hat{\tau}_{t-1}} \tau_t(\mathbf{X}_i) + \frac{\partial l(y_i, \hat{y}^{(t-1)})}{\partial \hat{f}_{t-1}} f_t(\mathbf{x}_i) + \frac{1}{2} \frac{\partial^2 l(y_i, \hat{y}^{(t-1)})}{\partial \hat{f}_{t-1}^2} f_t^2(\mathbf{X}_i)$$
$$+ \frac{1}{2} \frac{\partial^2 l(y_i, \hat{y}^{(t-1)})}{\partial \hat{\tau}_{t-1}^2} \tau_t^2(\mathbf{X}_i) + \frac{\partial^2 l(y_i, \hat{y}^{(t-1)})}{\partial \hat{f}_{t-1} \partial \hat{\tau}_{t-1}} f_t(\mathbf{X}_i) \tau_t(\mathbf{X}_i)]$$

Under the setting that $w_i \in [0, 1]$, we can remove the constant terms to obtain the following simplified objective at step $t$:

$$\tilde{\mathcal{L}}^{(t)} = \sum_{i=1}^n [w_i g_i \tau_t(\mathbf{X}_i) + \frac{1}{2} w_i h_i \tau_t^2(\mathbf{X}_i) + g_i f_t(\mathbf{X}_i) + \frac{1}{2} h_i f_t^2(\mathbf{X}_i) + w_i h_i \tau_t(\mathbf{X}_i) f_t(\mathbf{X}_i)]$$

where $g_i = \partial_{\hat{y}^{(t-1)}} l(y_i, \hat{y}^{(t-1)})$ and $h_i = \partial_{\hat{y}^{(t-1)}}^2 l(y_i, \hat{y}^{(t-1)})$ are first and second order gradient statistics on the loss function. Note that they are defined in the same way as standard gradient trees.

Define $I_j = \{i | q(\mathbf{X}_i) = j\}$ as the instance set of leaf $j$, we can rewrite the above equation as:

$$\tilde{\mathcal{L}}^{(t)} = \sum_{j=1}^T [(\sum_{i \in I_j} w_i g_i + w_i h_i f_j) \tau_j + \frac{1}{2} (\sum_{i \in I_j} w_i h_i) \tau_j^2 + (\sum_{i \in I_j} g_i) f_j + \frac{1}{2} (\sum_{i \in I_j} h_i) f_j^2]$$

We can further derive the optimal values for $f_j$ and $\tau_j$ of this dual quadratic function and the corresponding optimal weights $v^*$ and $u^*$. However, it is important to note that solving for both weights simultaneously will result in a significant decrease in the computing efficiency of the algorithm, compared to the standard regression tree, which has a simpler analytic solution for the quadratic function, during training process. This constrains the practical application of the scheme. We will introduce an approximation method to solve this difficulty in the next section.

## 5.2 Multi-objective Approximation

We point that if all $w_i$ are equal to 0, i.e., the data set contains only control samples, the above equation is identical to the optimization objective of the regression tree, and we can compute the optimal $v_0^*$ in that specific context. Under the setting that treatments are assigned randomly, we further assume that $v^* = v_0^*$ on each leaf, which enables us to derive the optimal weights $v^*$ with control instances. After that, the objective function degenerates to a simple quadratic function with one variable and we can solve optimal $u^*$. We can compute the optimal weights by

$$v_j^* = -\frac{\sum_{i \in I_j^0} g_i}{\sum_{i \in I_j^0} h_i}, \quad u_j^* = -\frac{\sum_{i \in I_j} w_i g_i + w_i h_i v_j^*}{\sum_{i \in I_j} w_i h_i} = -\frac{\sum_{i \in I_j^1} g_i + h_i v_j^*}{\sum_{i \in I_j^1} h_i}$$

where $I_j^0 = \{i|q(\mathbf{X}_i) = j, w_i = 0\}$ is the control instance set and $I_j^1 = \{i|q(\mathbf{X}_i) = j, w_i = 1\}$ is the treated instance set of leaf $j$. It is obvious that, after obtaining $v^*$ from the control group, $u^*$ is only related to the treated samples. This innovation simplifies the original solution process to sequentially solving two quadratic equations in one variable. It also enables the CausalGBM algorithm to scale to multiple treatment scenarios with minimal additional computational resources, as $u^*$ is computed based on the samples of the corresponding group independently of other groups. We then calculate the corresponding optimal loss by:

$$\tilde{\mathcal{L}}_{global}^{(t)}(q) = \sum_{j=1}^{T}[(\sum_{i\in I_j} g_i)v_j^* + \frac{1}{2}(\sum_{i\in I_j} h_i)(v_j^*)^2 - \frac{(\sum_{i\in I_j^1} g_i + h_i v_j^*)^2}{2\sum_{i\in I_j^1} h_i}]$$

Note that the value is obtained by computing all instances on leaf $I_j$ to ensure the global loss is optimized under this approximation method. We provide two other different forms of the optimum value, which are defined as:

$$\tilde{\mathcal{L}}_{local}^{(t)} = \sum_{j=1}^{T}[(\sum_{i\in I_j^1} g_i)v_j^* + \frac{1}{2}(\sum_{i\in I_j^1} h_i)(v_j^*)^2 - \frac{(\sum_{i\in I_j^1} g_i + h_i v_j^*)^2}{2\sum_{i\in I_j^1} h_i}]$$

and

$$\tilde{\mathcal{L}}_{\tau}^{(t)} = \sum_{j=1}^{T} - \frac{(\sum_{i\in I_j^1} g_i + h_i v_j^*)^2}{2\sum_{i\in I_j^1} h_i}$$

where $I_j^1 = \{i|q(\mathbf{X}_i) = j, w_i = 1\}$ is the treated instance set of leaf $j$. The first form is only influenced by the treatment group sample, and the other only by the causal effect. We present comparisons of the impact of the different forms for the model in the experimental section.

### 5.3 Greedy Algorithm for Tree Construction

Enumerating all possible tree structures to find the minimum loss is infeasible due to the combinatorial explosion. Instead, we employ a greedy algorithm that recursively bifurcates nodes, starting with a single parent node. We define the loss of a leaf as:

$$\tilde{\mathcal{L}}_{leaf}^{(t)} = (\sum_{i\in I_{leaf}} g_i)f_j^* + \frac{1}{2}(\sum_{i\in I_{leaf}} h_i)(f_j^*)^2 - \frac{(\sum_{i\in I_{leaf}^1} g_i + h_i f_j^*)^2}{2\sum_{i\in I_{leaf}^1} h_i}$$

Assume that $I_L$ and $I_R$ are the instance sets of left and right nodes after the split. Letting $I = I_L \cup I_R$, then the loss reduction after the split is given by:

$$\tilde{\mathcal{L}}_{split} = \tilde{\mathcal{L}}_I^{(t)} - (\tilde{\mathcal{L}}_L^{(t)} + \tilde{\mathcal{L}}_R^{(t)})$$

The above function will be used to evaluate the candidate split points. Compared with the xgboost algorithm, CausalGBM redefines the computation formulas for weights and evaluation functions in uplift modeling problems. As for the construction of the tree, we follow the computational framework of xgboost and lightgbm but have adjusted the calculation methods pertaining to weights and evaluations.

## 6 EXPERIMENTS

To comprehensively evaluate our proposed methods, extensive experiments were conducted on three large-scale real-world datasets and a synthetic dataset. These experiments aim to address the following research questions:

- **Q1**: What is the comparative performance of TDDP and CausalGBM against the baseline methods?

- **Q2**: What is the impact of switching the ensemble strategy from boosting to bagging on model performance?

- **Q3**: Is the performance of CausalGBM sensitive to different loss computation strategies?

### 6.1 Experiment Setup

#### 6.1.1 Datasets

Table 1: The basic statistics of datasets used in the paper.

| Metrics | CRITEO-UPLIFT$_{1M}$ | HILLSTROM | VOUCHER-UPLIFT | SYNTHETIC$_m$ |
|---|---|---|---|---|
| Size | 1,000,000 | 42,693 | 371,730 | 200,000 |
| Features | 12 | 8 | 2076 | $m$ |
| Avg. Label | 0.047 | 0.129 | 0.356 | 0.600 |
| Treatment Ratio | 0.85 | 0.50 | 0.85 | 0.50 |
| Relative Avg. Uplift (%) | 26.7 | 42.6 | 2.0 | 50.0 |

We evaluate our proposed models on three large-scale real-world datasets and a synthesized dataset. A summary of these datasets is given in Table 1. Here, we briefly introduce them as follows:

- HILLSTROM(Hillstrom, 2008): This dataset comprises data of 64,000 customers who made their last purchase within the past twelve months. The customers were divided equally into two treatment groups and one control group. The first treatment group received a "Men's Advertisement Email," the second a "Women's Advertisement Email," and the control group received no email. We focus on the comparison between the Women's merchandise email and no email, aligning with previous studies.

- CRITEO-UPLIFT(Diemert et al., 2018): A publicly available dataset comprises 25 million instances, each with 11 anonymized features. It includes a treatment indicator (promotional email receipt) and binary labels for visit and conversion status. We sampled 1 million instances from this dataset, using visit status as the response variable.

- VOUCHER-UPLIFT: Compiled from an e-commerce platform, this dataset details the user response to reminder messages accompanied by in-account vouchers. The primary outcome of interest is voucher usage within the subsequent seven-day period. The dataset includes 371,730 instances, each with 2,076-dimensional sparse features, a binary treatment indicator, and a binary response variable.

- SYNTHETIC$_m$: In order to compare the performances of various methods under ideal data conditions, we adopted the synthesis method proposed in (Chen et al., 2020). The dataset contains $m$ variables, a binary indicator and a binary label. About 70% of the variables are associated with both potential outcomes and uplift, and the remaining variables are redundant. The dataset consisted of 200,000 samples, and 5% of the samples are randomly labeled.

#### 6.1.2 Baseline Methods

Our comparative analysis encompasses three categories of baseline methods:

**Methods Extending Existing Supervised Learning Models**. This category includes extensions of established supervised learning techniques for causal effect estimation. Using LightGBM as the foundational model, we compare five variants:

- **The Single-Model Approach**(Athey & Imbens, 2015) . This method uses the concatenation of treatment and covariates as the features, and $y$ as the target to train a supervised model.

- **The Two-Model Approach**(Radcliffe, 2007). The two-model approach trains two models on the control and treated subjects separately, and then uses the difference of the two predictions as the estimated conditional average causal effect or uplift.

- **Transformed outcome Approach**(Jaskowski & Jaroszewicz, 2012). The transformed outcome approach transforms the observed outcome $y$ to $y^*$ such that the causal effect equals the conditional expectation of the transformed outcome $y^*$.

- **X-Learner**(Künzel et al., 2019). This method includes two models for response estimation, two sub-models for imputed treatment effects estimation, and a propensity model. All models are trained separately without the parameters shared.

- **R-Learner**(Nie & Wager, 2021). The R-learner is designed to estimate heterogeneous treatment effects by transforming the estimation problem into a regression problem. It operates by first estimating the propensity of receiving treatment and the outcome model to adjust for biases, and then directly modeling the treatment effect as a function of covariates by minimizing the residual between observed outcomes and those predicted by the initial models.

**Deep Learning-Based Methods**. These methods can capture complex non-linear relationships and handle high-dimensional data. We selected two variants:

- **TARNet**(Shalit et al., 2017). TARNet estimates causal effect by learning two functions parameterized by two neural networks (similar to the two-model approach). Before learning the two functions, TARNet utilises a shared feature representation across the treated and control subjects.

- **CFRNet**(Shalit et al., 2017). CFRNet applies an additional loss to the TARNet that forces the learned distributions of the treated and control covariates to be closer to each other. This is measured by either the maximum mean discrepancy (MMD) or the Wasserstein distance.

**Tree-Based Methods**. These methods employ binary tree models for treatment effect estimation, with three ensemble-based approaches considered:

- **Uplift Random Forest**(Guelman et al., 2012). URF is an uplift ensemble-based method. It consists of two steps. Firstly, it randomly samples a bootstrap training dataset and randomly selects covariates from dataset at each iteration. Secondly, UpliftDT (Rzepakowski & Jaroszewicz, 2012) is built on every training dataset from the above step. The set of uplift trees is used to predict the uplift for a new subject by using the average predictions of all trees. We select four splitting criteria based on KL divergence, $\chi^2$ divergence, Euclidean and the difference of uplift (DDP) between the two leaves for decision trees.

- **Causal Forest**(Wager & Athey, 2018). CF is a random forest-like algorithm that directly estimates the treatment effect. This method uses the Causal Tree (CT) algorithm as its base learner, and constructs the forest from an ensemble of $k$ causal trees.

- **Uplift Bayesian Random Forest**(Rafla et al., 2023). A random forest variant utilizing Uplift Bayesian Decision Trees, selecting the most probable model given the data through a Bayesian framework.

### 6.1.3 Evaluation Protocols

We perform 10-fold cross-validation and use two widely used metrics in prior work, including area under the ROC curve (AUC) and Qini coefficient (Diemert et al., 2018; Gutierrez & Gérardy, 2017; Zhang et al., 2021) (normalized by prefect Qini score) for evaluation, and we perform a grid search for hyperparameters to search for an optimal parameter set that achieved the best performance on the validation dataset, which consisted of 25% of the training dataset in each fold. The parameters we grid-searched included combinations of maximum depth (3, 4, 5) and number of trees (25,50,100,150). Other hyperparameters in algorithms keep their default values. All neural networks of various deep models consist of 128 hidden units with 3 fully connected layers. L2 regularization is applied to mitigate overfitting with a coefficient of 0.01, and the learning rate is set to 0.001 without decay.

### 6.2 Overall Performance Comparison (Q1)

To verify that our proposed methods can make the uplift prediction model more accurate, we compare TDDP and CausalGBM with different types of baselines and show their prediction performance on four large-scale datasets in Table 2. Here, we summarize key observations and insights as follows:

Table 2: Model performance evaluated by Qini coefficient on four datasets with corresponding mean and standard error. "S-", "T-", "TO-" and "URF-" stands for instantiations of single-model, two-model, transformed outcome and uplift random forest approaches, respectively. Note that for Qini larger value is better. Best results of all methods are highlighted in boldface.

| Model | HILLSTROM | CRITEO-UPLIFT$_{1M}$ | VOUCHER-UPLIFT | SYNTHETIC$_{100}$ |
|---|---|---|---|---|
| | Qini Coefficient (*mean ± s.e.*) | | | |
| S-LGB | 0.0616 ± 0.0179 | 0.0933 ± 0.0160 | 0.0032 ± 0.0053 | 0.1812 ± 0.0029 |
| T-LGB | 0.0567 ± 0.0168 | 0.0900 ± 0.0178 | 0.0014 ± 0.0054 | 0.1831 ± 0.0024 |
| TO-LGB | 0.0377 ± 0.0200 | 0.0941 ± 0.0201 | 0.0048 ± 0.0061 | 0.1832 ± 0.0044 |
| X-Learner | 0.0619 ± 0.0150 | 0.0929 ± 0.0246 | 0.0029 ± 0.0072 | 0.1824 ± 0.0030 |
| R-Learner | 0.0621 ± 0.0213 | 0.0936 ± 0.0197 | 0.0033 ± 0.0071 | 0.1829 ± 0.0027 |
| TARNet | 0.0636 ± 0.0203 | 0.0935 ± 0.0109 | 0.0045 ± 0.0076 | 0.1803 ± 0.0049 |
| CFRNet$_{wass}$ | 0.0635 ± 0.0218 | 0.0909 ± 0.0171 | 0.0042 ± 0.0082 | 0.1829 ± 0.0035 |
| CFRNet$_{mmd}$ | 0.0629 ± 0.0257 | 0.0923 ± 0.0146 | 0.0047 ± 0.0073 | 0.1808 ± 0.0047 |
| Causal Forest | 0.0617 ± 0.0139 | 0.0933 ± 0.0113 | 0.0055 ± 0.0041 | 0.1395 ± 0.0039 |
| UB-RF | 0.0595 ± 0.0127 | 0.0959 ± 0.0185 | 0.0081 ± 0.0092 | 0.1775 ± 0.0051 |
| URF-Chi | 0.0623 ± 0.0167 | 0.0925 ± 0.0125 | 0.0062 ± 0.0073 | 0.1003 ± 0.0054 |
| URF-ED | 0.0613 ± 0.0183 | 0.0942 ± 0.0141 | 0.0070 ± 0.0064 | 0.1657 ± 0.0057 |
| URF-KL | 0.0605 ± 0.0160 | 0.0926 ± 0.0123 | 0.0060 ± 0.0055 | 0.1457 ± 0.0036 |
| URF-DDP | 0.0599 ± 0.0161 | 0.0938 ± 0.0144 | 0.0072 ± 0.0060 | 0.1661 ± 0.0054 |
| TDDP | 0.0576 ± 0.0123 | 0.0884 ± 0.0179 | 0.0088 ± 0.0059 | 0.1836 ± 0.0045 |
| CausalGBM | **0.0643 ± 0.0246** | **0.0971 ± 0.0142** | **0.0108 ± 0.0042** | **0.1863 ± 0.0039** |

- **CausalGBM's superior performance**: Our proposed model CausalGBM outperforms all different baseline methods across all datasets. Specifically, it achieves relative performance gains of 1.1%, 1.3%, 22.7%, and 1.5% on four datasets, respectively, comparing to the best baseline. Further, Qini reflects the model's ability to give high predicted probabilities of persuasion to those who are actually more likely to be persuaded, and the improvement implies that our proposed model more accurately finds the target population for which the treatment is effective.

- **CausalGBM's robustness across different scenarios**: Comparing the results between the four datasets with different scales, we find that many baseline models are not robust in different scenarios. For example, URF-based methods perform significantly better on the HILLSTROM and CRITEO-UPLIFT$_{1M}$ than on the SYNTHETIC$_{100}$. A plausible reason is that the first two datasets are much smaller than the last two datasets in terms of feature dimensions. As a result the model's fitting ability is not the key on the first two datasets. In contrast, CausalGBM achieves the best performance in all datasets, which demonstrated our model's robustness across different scenarios.

  On the VOUCHER-UPLIFT dataset, we observe that deep learning and partial metamodeling-type methods driven by fitting potential outcomes under different treatments are weaker than tree models that focus on the heterogeneity of causal effects. Recall that, as seen in Table 1, this dataset has the weakest significance of causal effects of all the datasets. This poses a challenge to methods that focus only on potential outcomes, i.e., the label distributions of the different groups are very similar, which would result in the individual causal effects predicted by this class of methods close to zero. URF-based methods at minimizing heterogeneous differences remain well performing. Since the model of CausalGBM is trained by simultaneously computing both potential outcomes and causal effects, it still performs robustly on the weak causal effect dataset compared to the approaches driven only by potential outcomes.

- **An analysis of the volatility for TDDP on different datasets**: Comparing TDDP and URF-DDP, the only difference between the two methods lies in the ensemble learning method, the former employs boosting, while the latter takes bagging. Comparing the performances of the two methods on four datasets, TDDP performs better on the dataset with high-dimensional features, while URF-DDP performs better on the dataset consisting of low-dimensional features. A plausible reason is

that TDDP overfits the data in datasets with low-dimensional features, meanwhile, URF-DDP does not fit the data in high-dimensional datasets adequately.

## 6.3 Ablation Study

We compare how the ensemble learning method and the loss form contribute to the predictive performance of the model in this section. In order to better visualize the experimental findings and to prevent the derivation of conclusions from misleading information, we select the synthetic dataset and divide 50% as the training set and the remaining part as the test.

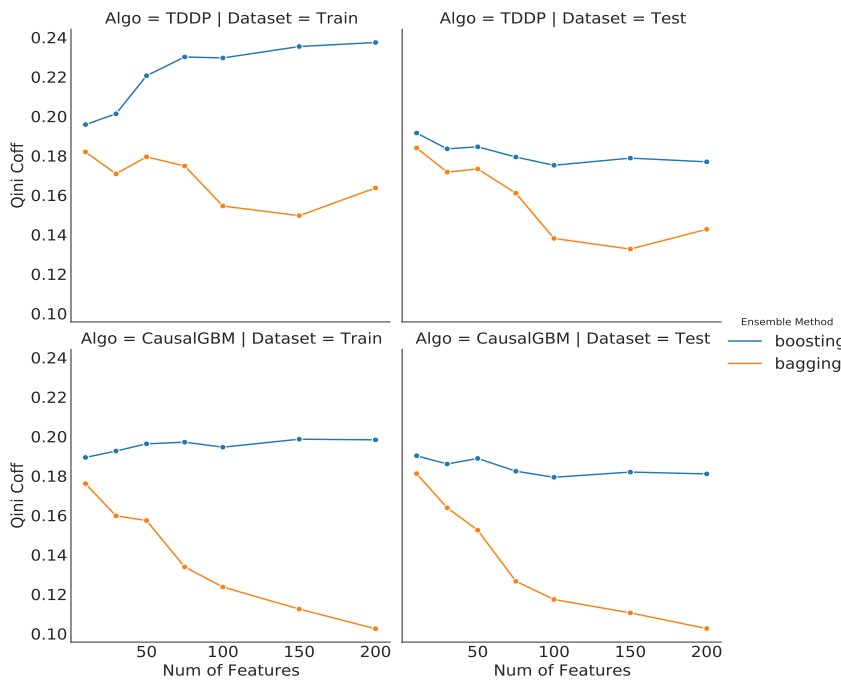

Figure 1: The ablation study on different ensemble methods.

### 6.3.1 How does the ensemble method facilitate the prediction performance? (Q2)

To answer this question, we compare the prediction results of TDDP and CausalGBM with its ablation version, which uses bagging as the ensemble mode. This blocks updating of the gradient for the CausalGBM algorithm, where the gradient is only computed before the training based on a uniform initial value. The gradient of each sample is constant throughout the training process. Overall, we observe that boosting significantly improves the model's capability to fit the training data compared to bagging across the two algorithms, and the gap widens as the feature dimensionality grows. Similar conclusions hold for the test dataset. This suggests that boosting provides a significant improvement in model performance, and that this approach is particularly suitable for high-dimensional datasets. On the low-dimensional dataset, the difference between the two methods is relatively small. In addition, we observe that the boosting version of the TDDP method is more likely to overfit the training data with increasing feature size than CausalGBM, leading to a weak generalization performance. In light of this finding, we suggest that tree models, which focus on the heterogeneity of local causal effects, require regularization methods to avoid overfitting, compared to the GBM approach that optimizes the global loss function.

### 6.3.2 Is the performance of CausalGBM sensitive to different loss computation strategies? (Q3)

We compare the performance of the models under different forms of loss. On the training dataset, $\tilde{\mathcal{L}}_\tau$, which focuses only on causal effects, shows superiority in terms of optimization speed and optimal score,

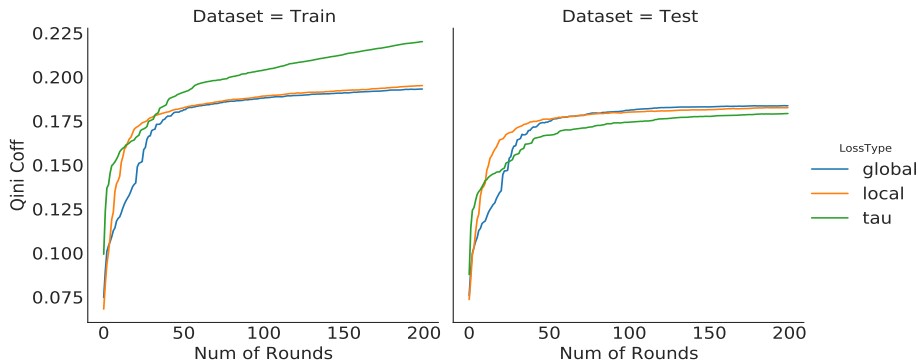

Figure 2: The ablation study on different loss types of CausalGBM.

significantly outperforming the other two formulations. $\tilde{\mathcal{L}}_{local}$ significantly outperforms $\tilde{\mathcal{L}}_{global}$ in terms of initial optimization speed, and both forms perform similarly in the middle and late stages of training. However, on the test dataset, a contrasting pattern emerges. $\tilde{\mathcal{L}}_{\tau}$ significantly underperforms the other two formulations in the middle stage. After 200 iterations, $\tilde{\mathcal{L}}_{global}$ achieves similarity to $\tilde{\mathcal{L}}_{local}$ in terms of the Qini Coefficient, with both surpassing the performance of $\tilde{\mathcal{L}}_{\tau}$.

We emphasize that if we focus only on the contribution of the causal effect prediction to the loss, the model will not be able to determine whether the prediction of the potential outcome is accurate. Since the causal effects are computed after the potential outcomes are obtained, any bias in the prediction of the outcomes will be transferred to the estimation of the causal effects. This invariably impacts model's generalization performance. Regarding $\tilde{\mathcal{L}}_{local}$, we highlight that excluding the control group is equivalent to increasing the contribution of the causal effect to the loss, which will force the model to pay more attention to it. There is no significant difference between the two in terms of final predictive performance. However, $\tilde{\mathcal{L}}_{local}$ may outperform $\tilde{\mathcal{L}}_{global}$ in situations where the sample size of the experimental group is much smaller than that of the control group. In such case, the loss value will come predominantly from the control samples. By employing $\tilde{\mathcal{L}}_{local}$, the model effectively mitigates the risk of disregarding causal effects.

## 7 Conclusion

In this paper, we formulate two novel boosting methods for the uplift modeling problem. The first algorithm we propose follows the idea of maximizing the heterogeneity of causal effects. This algorithm differs from the standard regression tree, because there is no loss function for which we are evaluating the gradient at each step, in order to minimize this loss. In contrast, the second algorithm we proposed, CausalGBM, fits both potential outcomes and causal effects by optimizing the loss function. This is similar to standard supervised learning methods. We demonstrate that our proposed techniques outperform the baseline model on large-scale real datasets, where the CausalGBM algorithm shows excellent robustness, while the TDDP algorithm needs to blend in some regularization methods to prevent the model from overfitting the training data. Our newly proposed algorithms are integrated into the UTBoost package, which is both open-source and ready-to-use.

Furthermore, both algorithms we propose can be directly extended to multi-treatment problems, and we will continue to explore the applicability of the algorithms with appropriate datasets in the future.

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

# A  Appendix

## A.1  Qini Coefficient

The Qini Curve is a metric designed to assess the quality of ranking based on the estimated uplift value. An effective uplift model should prioritize individuals more likely to respond to treatment, thereby achieving higher uplift values at the beginning of the curve. Despite its widespread use, variations in the definition of the Qini Curve exist within the literature, particularly regarding the plotted values and the derivation of a singular measure like the area under the curve.

To standardize the metrics for evaluation, consider a dataset $\mathcal{D} = \{(\mathbf{X}, y, w)\}$ consisting of $N$ instances, where $X$ denotes the feature vector, $y$ is a binary variable indicating response, and $w$ is a binary variable indicating treatment status. Given a predictive model $\hat{u}$, let $\pi$ represent the dataset's ordering in descending order of $\hat{u}$, such that $\pi(\mathcal{D}, k) = \{(\mathbf{X}_i, y_i, w_i) \in \mathcal{D}\}_{i=1,\ldots,k}$ includes the top $k$ instances according to this ordering, ensuring $\hat{u}(\mathbf{X}_i) \geq \hat{u}(\mathbf{X}_m)$ for all $i \leq k < m$. To quantify prediction performance, we define the total number and positive outcomes of treated and control instances among the top-$k$ ranked instances as follows:

$$N_\pi^T(\mathcal{D}, k) = \sum_{(\mathbf{X}_i, y_i, w_i) \in \pi(\mathcal{D}, k)} w_i$$

$$N_\pi^C(\mathcal{D}, k) = \sum_{(\mathbf{X}_i, y_i, w_i) \in \pi(\mathcal{D}, k)} (1 - w_i)$$

$$R_\pi^T(\mathcal{D}, k) = \sum_{(\mathbf{X}_i, y_i, w_i) \in \pi(\mathcal{D}, k)} y_i w_i$$

$$R_\pi^C(\mathcal{D}, k) = \sum_{(\mathbf{X}_i, y_i, w_i) \in \pi(\mathcal{D}, k)} y_i (1 - w_i)$$

The Qini Curve values are calculated by:

$$V_\pi(k) = R_\pi^T(\mathcal{D}, k) - R_\pi^C(\mathcal{D}, k) \frac{N_\pi^T(\mathcal{D}, k)}{N_\pi^C(\mathcal{D}, k)}$$

A baseline curve representing random ordering is introduced for comparison, with baseline values defined as:

$$V(k) = \frac{k}{n}(Y^T - Y^C \frac{N^T}{N^C})$$

Here, $Y^T$ ($Y^C$) and $N^T$ ($N^C$) denote the number of treated (control) instances responding and the total number of treated (control) observations in $\mathcal{D}$, respectively. A perfect model would assign higher scores to all treated individuals with positive outcomes over control individuals with positive outcomes. In this context, let $u^*(\mathbf{X}_i) = y_i w_i - y_i(1 - w_i)$ represent the optimal model, and $\pi^*$ denote the optimal ordering of the dataset

by $u^*$. Following Diemert et al. (2018), the Qini coefficient ($Q_\pi$) is defined as the ratio of the actual uplift gains curve above the baseline to that of the optimal Qini curve:

$$Q_\pi = \frac{\int V_\pi(k) - V(k)dk}{\int V_{\pi^*}(k) - V(k)dk}$$

We use a standard python package scikit-uplift[1] to compute this metric.

---

[1] https://github.com/maks-sh/scikit-uplift

