# OpenReview forum: "UTBoost: A Tree-boosting based System for Uplift Modeling"
_TMLR — Rejected by TMLR_

### Review · Reviewer_Y8X3 · 2024-03-12

**Summary Of Contributions:**

This paper introduces two innovative boosting techniques aimed at improving uplift modeling. The first method emphasizes maximizing the heterogeneity of causal effects and operates differently from standard regression trees by not using a loss function gradient evaluation at each step. The second method, named CausalGBM, aligns more with traditional supervised learning approaches by optimizing a loss function to fit potential outcomes and causal effects.

**Audience:**

Yes

**Broader Impact Concerns:**

NO.

**Claims And Evidence:**

Yes

**Requested Changes:**

Add experiments for comparing with the latest methods.

**Strengths And Weaknesses:**

Strengths:

1. The paper is easy to follow.

2. The motivation is clear.


Weaknesses:

My major concern is that the baselines are too old. Therefore, I am not very certain of the significance of the proposed method compared with the latest methods.

---

> ### Author Response · Authors · 2024-03-18
> **Reply to Reviewer Y8X3**
>
> Thank you for the insightful feedback, which is meaningful for improving the significance of the method we have proposed.
>
> We have included experimental comparison results for two baseline models. The first is the well-recognized R-Learner method [1], implemented by [2], which is a type of Meta-learning. The second is the UB-RF model [3], which, like our proposed method, falls within the tree model category and is implemented by [4]. This model uses a bagging approach for combining multiple models and applies the Bayesian evaluation criterion for uplift decision trees.
>
> We have tested both methods using the same experimental setup and have obtained the following results:
>
> |           | HILLSTROM             | CRITEO-UPLIFT         | VOUCHER-UPLIFT        | SYNTHETIC$_{100}$     |
> | --------- | --------------------- | --------------------- | --------------------- | --------------------- |
> | R-Learner | $ 0.0621 \pm 0.0213 $ | $ 0.0936 \pm 0.0197 $ | $ 0.0033 \pm 0.0071 $ | $ 0.1829 \pm 0.0027 $ |
> | UB-RF     | $ 0.0595 \pm 0.0127 $ | $ 0.0959 \pm 0.0185 $ | $ 0.0081 \pm 0.0092 $ | $ 0.1775 \pm 0.0051 $ |
>
> These results do not contradict the conclusions we reached in the experimental section of our paper.
>
> We are grateful for the opportunity to improve our work based on your feedback. We hope these additional results will satisfactorily address your comments regarding the significance of the proposed method.
>
>
>
> **References**
>
> [1] Nie X, Wager S. Quasi-oracle estimation of heterogeneous treatment effects[J]. Biometrika, 2021, 108(2): 299-319.
>
> [2] Chen H, Harinen T, Lee J Y, et al. Causalml: Python package for causal machine learning[J]. arXiv preprint arXiv:2002.11631, 2020.
>
> [3] Rafla M, Voisine N, Crémilleux B. Parameter-Free Bayesian Decision Trees for Uplift Modeling[C]. Pacific-Asia Conference on Knowledge Discovery and Data Mining. 2023: 309-321.
>
> [4] https://github.com/UData-Orange/kuplift

---

### Review · Reviewer_kqi4 · 2024-03-15

**Summary Of Contributions:**

In the paper, the authors proposed a method for learning boosted trees to predict treatment effects.
Under the assumption that the input $x$ and intervention $w$ are independent, the authors train the model using boosting, where decision trees are sequentially learned.
Two methods for learning decision trees are proposed: one minimizes the squared error with the treatment effect, and the other learns the output for each case of intervention presence or absence.

**Audience:**

Yes

**Broader Impact Concerns:**

There is no ethical concern.

**Claims And Evidence:**

Yes

**Requested Changes:**

1. By assuming independence between the input $x$ and intervention $w$, the problem addressed in this study becomes trivial. In this case, it should be sufficient to independently learn models to predict $y(0)$ from the dataset where $w=0$ and $y(1)$ from the dataset where $w=1$. It is necessary to discuss why there is a need for complex methods like the proposed approach and what the advantages of the proposed method are.

2. As demonstrated in Weakness 2, we can derive analytical updates for CausalGBM similar to conventional GBDT with little modification. What are the advantages of the heuristics used in the proposed method?

**Strengths And Weaknesses:**

### Strength
The proposed method demonstrated a better performance in predicting treatment effects compared to conventional methods in the experiments.

### Weakness 1
A critical issue in this study is the assumption of independence between the input $x$ and intervention $w$.
This assumes a Randomized Controlled Trial, where generally no particularly difficult processing is required.
If I understand correctly, in this case, it is sufficient to independently learn models to predict $y(0)$ from the dataset where $w=0$ and $y(1)$ from the dataset where $w=1$.
Generally, the difficulty in estimating treatment effects arises from the correlation between the input $x$ and intervention $w$.
By assuming independence, the problem addressed in this study becomes trivial.

### Weakness 2
The authors introduced heuristics for optimizing $\tau_j$ and $f_j$ in the learning of CausalGBM, but this seems not unnecessary.
In fact, by dividing the dataset into two sets: one with $w=0$ and the other with $w=1$, the following transformation becomes possible.

First, we define $\hat{y}_i$ as follows:
$$
\hat{y}\_i = \sum\_{m=1}^M \left( w\_i f\_m^1(X\_i) + (1 - w\_i) f\_m^0(X\_i) \right)
$$

Then, the objective function at the $t$-th step can be decomposed as follows:

$$
\mathcal{L}^{(t)} = \sum\_{i=1}^n \ell(y\_i, \hat{y}\_i^{(t-1)} + w\_i f\_t^1(x\_i) + (1 - w\_i) f\_t^0(x\_i)) = \sum\_{i: w\_i = 1} \ell(y\_i, \hat{y}\_i^{(t-1)} + f\_t^1(x\_i)) + \sum\_{i: w\_i = 0} \ell(y\_i, \hat{y}\_i^{(t-1)} + f\_t^0(x\_i))
$$

Then, by applying a second-order approximation to each term, we can obtain the following:

$$
\mathcal{L}^{(t)} \approx \sum\_{i: w\_i = 1} \left(\ell(y\_i, \hat{y}\_i^{(t-1)}) + g\_i f\_t^1(x\_i) + \frac{1}{2} h\_i f\_t^1(x\_i)^2 \right) + \sum\_{i: w\_i = 0} \left(\ell(y\_i, \hat{y}\_i^{(t-1)}) + g\_i f\_t^0(x\_i) + \frac{1}{2} h\_i f\_t^0(x\_i)^2 \right)
$$

These two terms are independent, and we can obtain analytical forms of $f_t^1, f_t^0$ following the usual derivation of GBDT.

---

> ### Author Response · Authors · 2024-03-18
> **Reply to Reviewer kqi4**
>
> Thank you for the insightful feedback. Below, we aim to address the reviewer's concerns and remarks.
>
>
> Assuming that $x$ and $w$ are independent, the theoretical proof you provided is elegant and correct, and it aligns with the classical baseline framework known as TwoModel. This method constructs separate predictive models for the treatment and control groups; however, it still has some limitations in practice:
>
> 1. **Sample Size Reduction**: Since the data is split into treatment and control groups, each model is trained on only half of the data. This can be especially problematic for smaller datasets. If $N$ is the total number of samples, each model is trained on approximately $N/2$ samples, potentially reducing the predictive power of each model. At the same time, a smaller sample size also implies a higher risk of overfitting.
> 2. **Increased Complexity and Computational Cost**: Maintaining two separate models for the treatment and control groups increases the complexity of the modeling process. This can lead to higher computational costs and more time spent on model tuning and validation.
> 3. **Loss of Information**: Since the models are built separately, each model may not utilize potentially relevant information from the other group's data. This can lead to a loss of information that could have improved the uplift prediction.
>
> Furthermore, the TwoModel approach focuses on predicting the outcomes separately and can therefore overlook the "weaker" uplift signals. [1] illustrates this phenomenon in a simulation study. In contrast, our proposed CausalGBM framework makes the following improvements over the TwoModel method:
>
> 1. **Information Sharing Mechanism**: We define the learning objective as $( y(0) + w\tau )$ instead of the pair $(y(0), y(1))$. The motivation behind this is that it allows the prediction of $y(1)$ to share the results of $y(0)$, thus leveraging additional information from the control group samples, not just limited to the treatment group samples. Furthermore, as evident from the final form of loss $\tilde{\mathcal{L}}_{global}^{(t)}$, our algorithm computes splits by considering both treatment and control group samples simultaneously. This clearly avoids the issue of reduced sample size inherent in the TwoModel approach and, with a larger sample size, also reduces the risk of overfitting.
> 2. **Direct Modeling of Uplift**: An additional advantage of our proposed method is that it facilitates the direct modeling of uplift. In contrast to the TwoModel approach, which indirectly estimates uplift, our proposed method has a greater advantage when the causal effect is weak. We have demonstrated this phenomenon in the experimental section.
> 3. **Modeling Cost**: The CausalGBM algorithm is an end-to-end trained model. Clearly, TwoModel cannot resolve the Uplift prediction problem by training just one GBDT model. Reducing modeling costs is also valued by the machine learning community.
>
> We hope we have clarified the concerns of the reviewer. Please let us know of any remaining concerns!

---

> > ### Comment · Reviewer_kqi4 · 2024-04-17
> > **Re: Reply to Reviewer kqi4**
> >
> > I would like to thank the authors for the reply.
> > However, I have to say I can hardly agree with your points.
> >
> > >  independence between the input $x$ and intervention $w$
> >
> > First of all, I am not very sure if we could call the curret study as "uplift modeling."
> > The estimation of treatment effects is problematic when $x$ and $w$ are dependent.
> > The current study assumes independence, which is not a typical assumption and hence can be misleading to say "uplift modeling".
> >
> > > TwoModel
> >
> > It is OK to say that the proposed method *can* have some advantages **if there are some supporting evidences**.
> > However, the current study does not adopt TwoModel as the baseline.
> > Are there any evidences that support the reply?
> >
> > > "weaker" uplift signals
> >
> >  I do not see any difference of estimating $(y(0), \tau)$ and $(y(0), y(1))$.
> > Apparently, $\tau = y(1) - y(0)$ and hence estimating $\tau$ and estimating $y(1)$ are equivalent.
> > As I pointed out in my review, estimating $y(1)$ seems to be easier than estimating $\tau$.
> > Is the estimation of $\tau$ truly advantageous?

---

> > > ### Author Response · Authors · 2024-04-17
> > > **Response to Reviewer kqi4**
> > >
> > > Thank you for your feedback and questions. We respond to your questions below.
> > >
> > > > independence between the input $x$ and intervention
> > >
> > > Motivated by various application scenarios, there exist two closely related yet distinct research communities: the treatment effect heterogeneity modeling and the uplift modeling communities. Both communities aim to estimate the change in the outcome resulting from the alteration of the treatment for specific subjects. However, a key distinction lies in the algorithms developed by each community. The uplift modeling community's algorithms are inherently designed for data derived from **randomized experiments**, which ensures data independence. On the other hand, methods developed within the treatment effect heterogeneity modeling community can be used for both **experimental and observational data**.
> > >
> > > Given that our research exclusively relies on randomized experiments, the proposed method cannot be directly applied to observational data, thereby fitting within the realm of uplift modeling research. A comprehensive explanation of the connections and differences between them is provided by [1].
> > >
> > >
> > >
> > > > 1. TwoModel
> > > > 2. "weaker" uplift signals
> > >
> > > We have compared the performance of our proposed method with other baseline models in Table 2, which includes the Two-Model method, and we use the **T-LGB** to denote it.
> > >
> > > Our experimental findings align with several benchmarking studies [2, 3]. Despite the effectiveness of the Two-Model method has been validated, this method exhibits inferior performance compared to the uplift tree model which targets at $\tau$ on many datasets. This discrepancy is notably magnified in scenarios characterized by weak causal effects. [4] has identified several factors that contribute to the practical failure of the Two-Model approach in real-world situations. One of the factors for this is that the causal effect is too low compared to the natural conversion rate. Consequently, in cases of potential conflict or discrepancy, priority will tend to be given to the main effect.
> > >
> > > Driven by these insights, we try to include $\tau$ directly as one of the learning objectives in the CausalGBM method, so as to **strengthen the importance of $\tau$​ to the model**.
> > >
> > >
> > >
> > > **References**
> > >
> > > [1] Zhang W, Li J, Liu L. A unified survey of treatment effect heterogeneity modelling and uplift modelling[J]. ACM Computing Surveys (CSUR), 2021, 54(8): 1-36.
> > >
> > > [2] Devriendt F, Moldovan D, Verbeke W. A literature survey and experimental evaluation of the state-of-the-art in uplift modeling: A stepping stone toward the development of prescriptive analytics[J]. Big data, 2018, 6(1): 13-41.
> > >
> > > [3] Sołtys M, Jaroszewicz S, Rzepakowski P. Ensemble methods for uplift modeling[J]. Data mining and knowledge discovery, 2015, 29: 1531-1559.
> > >
> > > [4] Radcliffe N J, Surry P D. Real-world uplift modelling with significance-based uplift trees[J]. White Paper TR-2011-1, Stochastic Solutions, 2011: 1-33.

---

### Review · Reviewer_4UDn · 2024-04-07

**Summary Of Contributions:**

This work proposes a new tree-based boosting algorithm for uplift modeling.  To address uplift modeling in the presence of a treatment indicator, the authors utilize a boosting model and suggest a causal gradient boosting machine.  Finally, the authors provide experimental results in several benchmarks.

**Audience:**

No

**Broader Impact Concerns:**

I cannot find any discussion on the broader impacts of this work in the current manuscript.

**Claims And Evidence:**

No

**Requested Changes:**

- You need to use double quotation marks correctly.
- In Page 2, `we propose an approximate` should be `we propose an approximate`.
- In Page 2, `realworld` should be `real-world`.
- Could you provide the definition of $\bot$?
- Could you provide the detailed description of Assumptions 1 and 2?  What is the meaning of these assumptions?
- Why don’t we simply build a tree-based model conditioned on a treatment indicator?
- Why is the multi-objective approximation multi-objective?  Three objectives are summed up, right?  It seems like a single objective with three terms.
- Could you provide the definition of Qini coefficient?
- Could you provide the details of baseline methods?
- I cannot understand the main message of Figure 2.  They (global, local $tau$) seem all similar.
- There are many typos, grammatical errors, and misuse of latex syntax.  The authors should carefully revise the manuscript.

**Strengths And Weaknesses:**

**Strengths**

- It solves an uplift problem using a new tree-based boosting model.

**Weaknesses**

- Presentation and writing can be improved more.
- Technical contributions are unclear.
- Some details are missing.

---

### Public Comment · ~Tian-Zuo_Wang1 · 2024-03-08

Dear Action Editor,

I am fine to review it according to your suggested time! (I cannot respond to you directly, so I begin a new comment).

Best Regards,

---

### Decision · Action_Editor_vnDq · 2024-04-28

**Recommendation:** Reject

**Comment:**

I think the paper has merits in terms of the problem tackled and the proposed approach. However, I agree with reviewers 4UDn and  kqi4. Indeed, the proposed approach and the developed formulation provided are extremely simple with almost no modifications to the existing GBDT framework, while the complicated heuristic proposed by the authors is not shown to be advantageous over the current state of the art.
Furthermore, I have to agree with 4UDn when mentioning that the presentation, organization and exposure of the paper need to be significantly improved and the authors should make more clear which the main and relevant contributions they made in this manuscript. I also think that the cricism related to the assumption of independence between the input and the intervention, raised by kqi4, is relevant and the answer from the authors is not convincing or at least it does not solve the relevant issue, as also commented by kqi4. Finally, on this point the authors did not react to the following question from kqi4 "Is the estimation of tau truly advantageous?" and this  does not help to solve the issue or at least to clarify it.

**Audience:**

I personally think that the TMLR audience, even thought limited, could benefit from the topic and problem tackled by the paper to be presented and discussed. Therefore, from this point of view I agree with reviewers Y8X3 and kqi4 that stated the TMLR audience could benefit from this paper to be shared.

**Claims And Evidence:**

According to what reported by reviewers, we have that two of them concluded that claims are supported by evidence, while the third reviewer points out that the claims of the paper are not supported by the evidence provided by the authors. However, going through the paper and after the discussion which took place with reviewers I think that claims are supported, at least partially, by the evidence provided in the manuscript.

**Resubmission Of Major Revision:**

The authors may consider submitting a major revision at a later time.